# *Clostridium difficile* Infections in an Emergency Surgical Unit from North-East Romania

**DOI:** 10.3390/medicina59050830

**Published:** 2023-04-24

**Authors:** Bogdan Mihnea Ciuntu, Gheorghe G. Balan, Mihaela Buna-Arvinte, Irina Mihaela Abdulan, Adelina Papancea, Ștefan Lucian Toma, Bogdan Veliceasa, Oana Viola Bădulescu, Gabriela Ghiga, Ana Maria Fătu, Mihai Bogdan Vascu, Antonia Moldovanu, Dan Vintilă, Alin Mihai Vasilescu

**Affiliations:** 1Department of General Surgery, “Grigore T. Popa” University of Medicine and Pharmacy, Universitatii Street, No. 16, 700115 Iasi, Romania; bogdan-mihnea.ciuntu@umfiasi.ro (B.M.C.); buna.mihaela@umfiasi.ro (M.B.-A.); papancea.adelina@umfiasi.ro (A.P.); dan.vintila@yahoo.com (D.V.);; 2Department of Gastroenterology, “Grigore T. Popa” University of Medicine and Pharmacy, Universitatii Street, No. 16, 700115 Iasi, Romania; gheorghe-g-balan@umfiasi.ro; 3Department of Medical Specialties I, “Grigore T. Popa” University of Medicine and Pharmacy, 700115 Iasi, Romania; 4Department of Materials Engineering and Industrial Security, Faculty of Materials Science and Engineering, Gheorghe Asachi Technical University of Iasi, 700050 Iasi, Romania; stl_toma@yahoo.com; 5Department of Traumatology and Orthopaedics, Faculty of Medicine, “Grigore T. Popa” University of Medicine and Pharmacy, 700115 Iasi, Romania; bogdan.veliceasa@umfiasi.ro; 6Department of Haematholohy, Faculty of Medicine, “Grigore T. Popa” University of Medicine and Pharmacy, 700115 Iasi, Romania; viola.badulescu@umfiasi.ro; 7Department of Mother and Child Medicine, Faculty of Medicine, “Grigore T. Popa” University of Medicine and Pharmacy, 700115 Iasi, Romania; gabriela.ghica@umfiasi.ro; 8Department of Implantology Removable Denture Technology, Discipline of Ergonomy, Faculty of Medicine, “Grigore T. Popa” University of Medicine and Pharmacy, 700115 Iasi, Romania; ana.fatu@umfiasi.ro; 9Department of Odontology, Periodontology and Fixed Prosthesis, Faculty of Medicine, “Grigore T. Popa” University of Medicine and Pharmacy, 700115 Iasi, Romania; mihai.vascu@umfiasi.ro (M.B.V.); antonia.moldovanu@umfiasi.ro (A.M.)

**Keywords:** *Clostridium difficile* infection, colitis, surgical intervention, mortality

## Abstract

*Background and Objectives*: Colitis with *Clostridium difficile* is an important health problem that occurs with an intensity that varies between mild and severe. Surgical interventions are required only in fulminant forms. There is little evidence regarding the best surgical intervention in these cases. *Materials and Methods*: Patients with *C. difficile* infection were identified from the two surgery clinics from the ‘Saint Spiridon’ Emergency Hospital Iași, Romania. Data regarding the presentation, indication for surgery, antibiotic therapy, type of toxins, and post-operative outcomes were collected over a 3-year period. *Results*: From a total of 12,432 patients admitted for emergency or elective surgery, 140 (1.12%) were diagnosed with *C. difficile* infection. The mortality rate was 14% (20 cases). Non-survivors had higher rates of lower-limb amputations, bowel resections, hepatectomy, and splenectomy. Additional surgery was necessary in 2.8% of cases because of the complications of *C. difficile* colitis. In three cases, terminal colostomy was performed and as well as one case with subtotal colectomy with ileostomy. All patients who required the second surgery died within the 30-day mortality period. *Conclusions*: In our prospective study, the incidence was increased both in cases of patients with interventions on the colon and in those requiring limb amputations. Surgical interventions are rarely required in patients with *C. difficile* colitis.

## 1. Introduction

*Clostridium difficile* is an anaerobic, sporulated Gram-positive bacillus that was discovered in 1935. Considered a rare infection until 1970, after the introduction of antibiotic treatment, the *C. difficile* infection rate increased [1]. Studies have shown that approximately 5% of adults and 15–70% of children are colonized with *C. difficile*. However, only a small portion develops the infection, being protected by normal intestinal peristalsis and an intact microbial intestinal flora [2]. Consequently, two entities of the pathology are defined: colonization (detection of bacteria without symptoms predominantly nosocomial infection) and infection (detection of toxins and symptoms) [3].

It is highly transmissible through the fecal–oral route and its exotoxins cause a spectrum of disease ranging from mild diarrhea to severe complications such as dehydration, infectious colitis, toxic megacolon, colonic perforation, sepsis, circulatory shock, and death [4]. The mortality rate increases with the severity of the infection, reaching around 34% in cases of patients admitted to the ICU [5]. 

As it was mentioned before, although *C. difficile* toxins were highlighted earlier by researchers, it was only in 1970 that their role in the development of pseudomembranous colitis was confirmed. This bacterium produces at least two toxins called A (TcdA) and B (TcdB). Toxin A, an enterotoxin, causes an increase in fluid secretion and inflammation in the intestinal mucosa. Toxin B, which in vitro exhibits a cytotoxic action approximately 1000 times stronger than toxin A, acts synergistically with it. While toxin A is produced by almost all strains of *C. difficile* involved in the disease, it has recently been shown that some strains secrete only toxin B [1].

In 2002, a variant of this bacterium was discovered, ribotype 027, initially in North America, later in other parts of the world. The epidemiological importance comes from the fact that it is more aggressive and contagious by synthesizing an additional toxin [6]. Until 2011, the incidence of *C. difficile* infection also increased in Romania, which correlated with the highest share determined by ribotype 027 recorded in the European Union states, approximately 70% of all cases [7]. 

In a country report from 2018 that analyzed the evolution of *C. difficile* infection in Romania, out of a total of 10,241 confirmed cases, 7743% (76%) were classified as healthcare-associated infections, 1931 (19%) as community infections and 567 (6%) were infections of undetermined origin [8].

Patients at highest risk for *C. difficile* infection include hospitalized individuals aged over 65 years old with recent antibiotic exposure. Pertinent explanations include depletion of protective gut flora by antibiotics and a diminished immune response to *C. difficile* due to age and medical comorbidities [9]. Most epidemics occur in the hospital setting and in long-term care facilities, but outpatient acquisition is also described [10]. The risk factors of community-acquired infections, apart from those mentioned before, are white race, cardiac disease, chronic kidney disease, and inflammatory bowel disease [11]. 

Active monitoring and limiting the unnecessary administration of antibiotics are key to minimizing this type of infection. The incidence rates obtained from a standard surveillance system can be used as an important indicator of the quality of healthcare. In Europe, epidemiological data on *C. difficile* infection in acute care units are derived from a few limited studies, with significant differences in study design [12].

However, antibiotic treatment cannot always resolve the effects of the infection. The surgical option is indicated in complicated *C. difficile* infection because of fulminant colitis, toxic megacolon, severe ileus, or colonic perforations. The outcome is reserved with a high mortality [13], but surgery performed before the onset of multi-system organ failure (MSOF) and hemodynamic instability could increase the rate of survival [14]. 

The aim of this study was to find out an updated rate of *C. difficile* infections in a large surgical clinic from the north-east of Romania considering the extent of empiric antibiotic treatments nowadays. The specifics of the clinic mainly include acute and trauma cases. At the same time, we wanted to observe if there were differences in the evolution of patients depending on the pathologies that required hospitalization, the operations performed, and the medical treatment administered, considering the fact that some pathologies required pre- and post-operative antibiotic treatment.

## 2. Materials and Methods

### 2.1. Study Design and Setting

This prospective study was conducted between September 2019 and September 2022, in the First and Second Surgery Clinic from the Emergency Hospital ‘Saint Spiridon’ from Iasi, Romania. Our research included the patients admitted for emergency surgeries. 

### 2.2. Study Participants

From a total of 12,432 patients that were admitted to our clinic, we identified 140 patients who were diagnosed with post-operative *C. difficile* infection within the same hospital stay. Data regarding the age, sex, diagnostic of admission, type of surgery performed, antibiotic therapy used, type of toxins, evolution, and post-operative outcomes were collected.

### 2.3. Diagnosis and Laboratory Technique

In order to confirm the infection, samples were taken from freshly emitted feces in single-use containers without preservative. They were processed within a safety interval of 3 h in an external certified laboratory. 

A standard amount of the diluted sample was mixed with conjugate solution 1 (contained specific monoclonal antibodies against *C. difficile* toxin A and toxin B conjugated with colored microparticles) as well as with conjugate solution 2 (contained specific anti-toxin A and anti-toxin B antibodies biotinylated).

A volume of this mixture was transferred into a dedicated window of the test box; the box contained immobilized streptavidin in the test strip and goat anti-immunoglobulin antibodies in the control strip. Specific antibodies bound to the antigen in the sample and form “sandwich” antigen–antibody complexes.

The complexes migrated through capillarity, reaching the area containing the test strip, binding to the streptavidin present in the solid phase and were visualized in the form of a black colored band, of any intensity, in the results window. In the absence of toxin A and/or B from the sample, no band was visualized. The test was validated only if a band colored black, of any intensity, was obtained in the control window.

After the diagnosis, the patients were isolated to prevent the spread of the infection. 

### 2.4. Ethical Approval

In order to be admitted to the study, all the patients completed an informed consent form. The ethics approval was received in 2018 (no. 8404/03.05.2018 issued by the Ministry of National Education—University of Medicine and Pharmacy Gr. T. Popa—Iasi).

### 2.5. Statistical Methods

Data analysis was performed using SPSS 20.0 (Statistical Package for the Social Sciences, Chicago, IL, USA). For continuous data, the distribution was assessed by the Shapiro–Wilk test. Data were entered as the mean ± standard deviation, or a number with a percent frequency for continuous variables with a normal distribution. Continuous variables with normal distributions were compared using independent samples for the Student’s *t*-test in the case of two samples. For continuous variables not satisfying the assumption of normality, the evaluation was done by applying nonparametric tests, i.e., the Mann–Whitney U test in the case of two samples. Results with a *p*-value < 0.05 were considered statistically significant.

## 3. Results

From a total of 12,432 patients admitted, 140 were diagnosed with *C. difficile* infection (1.12%). The average age of was 64.42 ± 16.31, 52.8% of them were men and 47.14 % women. A total of 20 (14.28%) patients died within the same hospital stay. The differences between survivors and non-survivors are illustrated in Table 1. 

From all the patients, 34.28% (*n* = 48) were contacts with at least one patient that was positive for *C. difficile* infection. Both toxins were identified after surgery in 82.14% (*n* = 115), with no significant differences between survivors and non-survivors.

Most of the patients, 85% (*n* = 120) were admitted directly to our clinic and 15% (*n* = 20) of cases were initially admitted in other medical specialties. Complicated forms of the infection were found in 17.85% (*n* = 25) due to multiple systems organ failure.

Neoplasm of the colon was the main diagnostis in 17.85% (*n* = 25), 10 % were acute cholecystitis, incisional hernia, and gangrene of the lower limb (Figure 1).

Regarding the election of the medical treatment, approximately two thirds received monotherapy (Vancomycin) and one third double had antibiotic therapy (vancomycin and metronidazole). The dosage we used for vancomycin was 125 mg PO q6hr and for metronidazole, 500 mg PO q8hr for 10 days. We noted a difference between treatments in the case of survivors (Table 2).

Analyzing the surgical interventions, there were significant differences between survivors and non-survivors. There were higher rates of lower-limb amputations, bowel resections, hepatectomy, and splenectomy in the survivors (Table 3).

Surgery was necessary in four patients (2.8%) because of the complications of *C. difficile* colitis, due to toxic megacolon and (*n* = 3) and bowel perforation (*n* = 1). 

In four of the cases, emergency surgical intervention was needed due to infection complications. Of these, in three cases, terminal colostomy was performed and in one case subtotal colectomy with ileostomy. All patients were admitted to the intensive care unit post-operatively with a mean stay of 10 days. 

The total mortality rate in *C. difficile* infections was 14% (*n* = 20), but all patients who required additional surgery died within the 30-day mortality period. 

## 4. Discussion

The present research is a prospective one in which we collected information over a 3-year period. From 12,432 patients admitted, 1.12% were diagnosed with *C. difficile* infection. Previous studies have reported an incidence of up to 7.8%, reaching three times higher values in cases of patients with colon surgery [15,16].

More than half of the deceased patients (60%) had a history of multiple hospitalizations, with statistically significant differences when compared to the surviving patients. The data obtained are consistent with the data from the literature, considering that healthcare exposures before the hospitalization may increase the risk for *C. difficile* infection [17].

Two large clostridial glucosylating toxins, toxin A (TcdA) and toxin B (TcdB), are involved in the pathogenesis of *C. difficile*. There are some strains that produce a third toxin, the binary toxin *C. difficile* transferase, which can also contribute to *C. difficile* virulence and disease. These toxins act on the colonic epithelium and immune cells and induce a complex cascade of cellular events that result in fluid secretion, inflammation, and tissue damage, which are the hallmark features of the disease. In our study, 82.14% presented both toxins, these being the majority both in the case of the survivors and non-survivors.

In a large-scale study that followed patients from 52 hospitals from Michigan, USA, between 2012 and 2013, Abdelsattar et al. showed that the highest rate of *C. difficile* infection occurred after the interventions of amputation of the lower limbs, bowel resection, and esophageal and gastric interventions [18]. In our study, the results were slightly reversed with the highest percentage being interventions on the colon followed by amputations and eso-gastric interventions.

Moreover, we observed statistically significant differences only in the case of limb amputation operations (30% versus 8.3%). It is worth mentioning that in cases of patients with a bad outcome, there was a higher rate of bowel resection, 35% compared to 30% in the case of survivors.

In the treatment approach to *C. difficile* infection, clinicians aim to cure both the first episode and to reduce the risk of recurrences. In the recently updated Infectious Diseases Society of America/Society for Healthcare Epidemiology of America (IDSA/SHEA) guidelines and the updated European Society of Clinical Microbiology and Infectious Diseases (ESCMID) guidance document, fidaxomicin is now recommended as the first treatment option over vancomycin for both the first episode and for relapse *C. difficile* infection. Although vancomycin is still a suitable alternative to fidaxomicin, specific focus was placed on the improved results and reduced risk of relapse observed with fidaxomicin compared to vancomycin [19,20]. Consequently, the new recommendations converge on a main idea: to consider fidaxomicin first, considering the global benefits of the patient, if feasible. Regarding the last aspect, fidaxomicin is more expensive than vancomycin, so accessibility remains reduced in multiple hospitals. ESCMID guide through a good practice statement mentions that when fidaxomicin is unavailable or unfeasible, vancomycin remains a suitable alternative.

Referring to the treatment options used in the surgery clinics from our study, the high cost of this macrolide makes it difficult to be used as the first option in the treatment of this type of infection for the time being. Consequently, the treatment options used in that period were metronidazole and vancomycin.

The choice of treatment—monotherapy or bitherapy—did not present differences between the two categories, survivors and non-survivors, that followed the treatments according to current recommendations [21]. However, in the case of survivors, monotherapy was preferred in 63.33% of cases (*p* = 0.000002), while in the case of those with a bad outcome, double association was chosen in 55% of cases.

Several studies have shown that a prophylactic dose given before surgery is associated with substantially reduced rates of wound infection and post-operative sepsis, especially in the case of bowel interventions [22,23]. Additionally, in these cases, the risk of developing *C. difficile* infection was outweighed by the benefit of avoiding a wound infection. It should be noted that there were also other pathologies/surgical interventions that required antibiotic treatment (gangrene of the lower limb) in the case of the patients included in our study.

Of the 140 patients, 37.85% had antibiotic treatment pre-operatively, and 85.71% after the surgical intervention, with a slightly higher rate in the case of patients with a bad outcome. There were no significant differences between the two groups. Currently, there is no consensus on the pre-operative administration of antibiotics, but a short course for bowel preparation is insisted upon, if necessary, since prolonged treatment can be a risk factor for developing *C. difficile* infection in patients undergoing colorectal surgery [24]. 

Post-operatively, the unfavorable evolution of the symptoms appeared in 17.85% of the cases. Multiple systems organ failure occurred in 5% of the survivors, the percentage reaching up to 95% in the case of patients who died. Factors such as advanced age, the presence of comorbidities, and prolonged previous treatment with PPIs can hinder the evolution of the infection, despite the initiation of prompt antibiotic treatment [25]. Several reported cases showed that even after using all the available resources, evolution can be lethal. In addition, when multiple system organ failure is present, survival is not positively influenced by radical intervention or subtotal colectomy [26,27].

*C. difficile* has become a significant public health threat in the past decade, largely due to the emergence/selection of hypervirulent strains that persist in healthcare-associated settings and cause more severe infections [28]. These strains are now being associated with disease in healthy individuals who are not part of the population considered to be at risk [29,30].

The presence of the infection and the growing aggressiveness of the pathogen influence the post-operative evolution of the patients. Moreover, some cases cannot be solved by antibiotic treatment and worsen, requiring emergency interventions, with extensive bowel resections, whose evolution is most often unfavorable. These cases especially appear in tertiary, large, university hospitals, where the number of performed surgeries has increased, but also the diversity and complexity of the cases are greater.

*C. difficile* infection occurs as a chronic or an acute illness with intensity varying from mild to severe. Most cases can be managed with antibiotics and supportive care. Surgical intervention in terms of colectomy is rarely required in patients with *C. difficile* colitis. However, when the patient presents with fulminant disease, the early decision to perform surgery is imperative for survival. Performed before the total morpho-functional degradation of the colon and the development of complications, surgery has lower rates of complications and post-operative lethality.

The current standard of care is subtotal colectomy. However, loop ileostomy with vancomycin enemas delivered into the colonic mucosa has been described as a viable option on selected patients [12]. 

In our study, 2.8% of the cases needed additional surgery because of the complications of *C. difficile* infection. The intervention rate is slightly higher compared to other studies, but the comparison is difficult considering the small groups studied in previous research [31]. 

It is important to emphasize that multidisciplinary collaboration and early surgical intervention can improve the outcome of patients with fulminant infection. 

Fulminant *C. difficile* colitis remains associated with exceptionally high mortality following surgical intervention. The adoption of loop ileostomy as a valid alternative to conventional surgical interventions, such as total colectomy, has more than doubled over the past few years. The data in this study corroborated prior findings regarding equivalent outcomes between both procedures. While the results from randomized clinical trials and a better understanding of functional outcomes are both needed, it appears that loop ileostomy is a viable alternative for acute care surgeons during management of fulminant colitis [32,33].

The strength of the study is that it is a prospective one. The limitations consist of the small batch of patients from a single center, the incomplete medical history, unclear previous treatment with empiric antibiotherapy, and self-prescribed PPI intake. 

## 5. Conclusions

*C. difficile* infection remains a major public health problem, especially in surgical wards. In our research, the incidence was increased both in the case of patients with interventions on the colon, and in those requiring limb amputations. The low number of patients who developed the infection is a positive aspect, but we are currently considering follow-up over a longer period of time to be able to include a larger number of patients, also from other regional surgical centers. Thus, we will be able to have an objective and more extensive perspective on the subject.

Clear and rigorous assessment of the risk factors for *C. difficile* infection is recommended to adjust pre-operative care and surgical management. Further studies are needed to complete our findings and a multidisciplinary approach is mandatory when it comes to proper prevention and treatment.

## Figures and Tables

**Figure 1 medicina-59-00830-f001:**
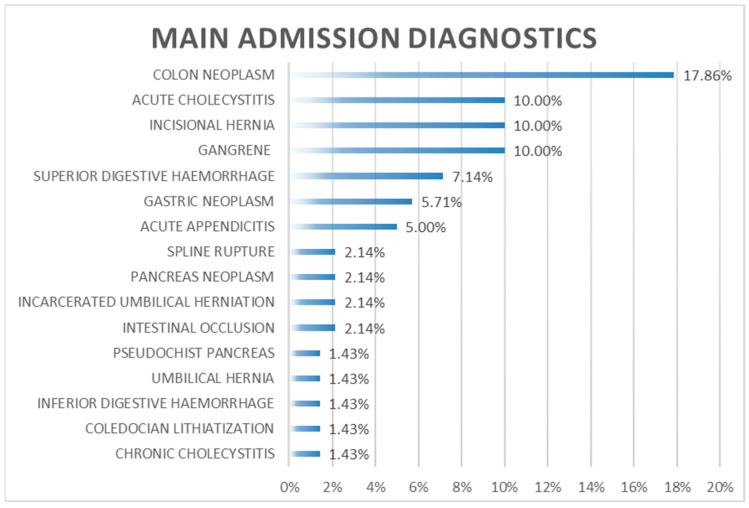
Main admission diagnostics.

**Table 1 medicina-59-00830-t001:** Characteristics of the study group and differences between the group of survivors and the non-survivors.

	Total *n* = 140	Survivors *n* = 120	Non-Survivors *n* = 20	*p*-Value
Age (mean ± SD)	64.42 ± 16.31	63.66 ± 15.64	68.95 ± 19.76	0.18
Gender, *n* (%)				
Men	74 (52.86)	69 (57.5)	5 (25)	0.14
Women	66 (47.14)	51 (42.5)	15 (75)	
*C. difficile* toxin, *n* (%)				
A	20 (14.28)	17 (14.16)	3 (15)	0.92
B	5 (3.57)	4 (3.33)	1 (0.5)	0.71
A + B	115 (82.14)	99 (82.5)	16 (80)	0.78
Contact with *C. difficile* infection, *n* (%)	48 (34.28)	41 (34.16)	7 (35)	0.94
Perioperative antibiotic therapy, *n* (%)	53 (37.85)	42 (35)	11 (55)	0.08
Post-operative antibiotic therapy, *n* (%)	120 (85.71)	103 (85.83)	17 (85)	0.92
Treatment, *n* (%)				
Vancomycin	85 (60.71)	76 (63.33)	9 (45)	0.12
Vancomycin + Metronidazole	55 (39.28)	44 (36.66)	11 (55)	
MSOF, *n* (%)	25 (17.85)	6 (5)	19 (95)	**<0.00001**
Previous multiple hospitalizations, *n* (%)	68 (48.57)	56 (46.66)	12 (60)	**0.038**

MSOF-Multiple systems organ failure.

**Table 2 medicina-59-00830-t002:** Election of the antibiotic therapy.

	Vancomycin	Vancomycin and Metronidazole	*p*-Value
Survivors, *n* (%)	76 (63.33)	44 (36.66)	0.000002
Non-survivors, *n* (%)	9 (45)	11 (55)	0.53

**Table 3 medicina-59-00830-t003:** Types of surgical interventions.

Type of Surgery	Total *n* = 140	Survivors *n* = 120	Non-Survivors *n* = 20	*p*-Value
Gastric or esophageal operations, *n* (%)	21 (15)	21 (17.5)	0 (0)	-
Bowel resection or repair, *n* (%)	47 (33.57)	40 (30)	7 (35)	0.78
Hepatectomy, *n* (%)	2 (1.4)	1 (0.83)	1 (5)	0.56
Splenectomy, *n* (%)	3 (2.14)	2 (1.66)	1 (5)	0.14
Pancreatectomy, *n* (%)	5 (3.57)	5 (4.16)	0 (0)	-
Cholecystectomy, *n* (%)	19 (13.57)	18 (15)	1 (5)	0.22
Lower-extremity amputation, *n* (%)	16 (11.42)	10 (8.3)	6 (30)	**0.04**
Hernia, *n* (%)	19 (13.57)	19 (15.8)	0 (0)	-
Other, *n* (%)	4 (2.85)	4 (3.33)	0 (0)	-

## Data Availability

The data published in this research are available on request from the first author and corresponding authors.

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
