# Peer review of "Clostridium difficile Infections in an Emergency Surgical Unit from North-East Romania"

_medicina, 2023, doi:10.3390/medicina59050830_

Round 1

Reviewer 1 Report

Topic

The manuscript “Clostridium Difficile infections in an emergency surgical unit

from North-East Romania”

Findings

The researchers investigated the Clostridium difficile infection, prospective study, collected data for 3 years, clinical and surgical data. Data were collected from a hospital in Romania,

140 patients diagnosed with CD infection were included in the analysis, with mortality rate o 14%,

Higher rate of lower limb amputations, bowel resections, hepatectomy and splenectomy and addition surgery among non survivors. Type of surgery was terminal colostomy in 3 cases and subtotal colectomy with ileostomy in one case. The authors concluded that surgical interventions are rarely required in patients with CD colitis.

The strength of the study that it is a prospective study, if it is really prospective, because in the manuscript was prospective/retrospective.

The limitations of the study, still small number of patients, in a single center.

The are several points should be improved:

1.      Is this a retrospective or prospective study?

Line 78 the authors wrote prospective study, while in the discussion, line 157 the author wrote retrospective study. non mention of the study design in the abstract.

2.      I suggest adding the number of patients who had CD infection and the number of deaths in the abstract and not only the percentage.

3.      Suggest to calculate the risk of non-mortality in a multivariate analysis.  

4.      The conclusions are too long and not really conclusions, most of it could be part of the introduction or discussion. Conclusions should be short with clear messages/ conclusions.

Reviewer 2 Report

The article presents data about morbidity caused by a very important bacteria. The article deals with complications after surgery, so it is very important to publish the results. Especially as there is reference to mortality and the treatment the patients received. It is a shame that there is no reference to the molecular aspect of the bacteria, i.e. to which ST the bacteria belong

Reviewer 3 Report

Dear Authors,

Thank you for the opportunity to review your work. It seems interesting to me, but in my opinion it needs a major revision.

1.       According to the academic rules, the name Clostridium difficile is always written in italic. The whole work needs editorial improvement in this point.

2.       I suggest resigning from the abbreviation CD in reference to Clostridium difficile. Customarily "CD" it is reserved for coeliac disease or Crohn's disease. The abbreviation C. difficile is usually used in publications.

3.       There is a mess in entering shortcuts. Line 41 introduces the abbreviation "Clostridium difficile infection (CDI)" while in line 46 is "CD infection". It is similar in other places of work, e.g.: verse 174.

4.       Introduction: There is no information about the prevalence of C. difficile in the environment, and especially it is a fairly common bacterium. The information about the difference between colonization and infection is correct. Similarly, the article introduction lacks information on C. difficile toxins (toxin A (TcdA) and toxin B (TcdB), they are in the discussion. I suggest that this information be expanded, e.g. in line 55.

5.       Material and methods: What "laboratory technique used was the immunochromatography" (line 88) was used in the Laboratory. The name of the exact test, manufacturer and equipment should be listed. Was this technique used to confirm infection or presence of C. difficile toxins? This information is missing.

6.       Results:

a.        Table 1 lacks information on what the p-value refers to. The reviewer guesses, and it should be written.

b.       Please explain why the Multiple systems organ failure information in line 119 is different from table 1

7.        "Materials and methods" or in  "Result" lack information on the dosage of used antibiotics.

8.       Discussion: The opening lines of the discussion (147-156) should be included in the introduction or rejected because do not add anything to this section.

9. Conclusions: According to the reviewer, this part is too general and does not refer to the conclusions drawn from the work. It should be corrected.

Round 2

Reviewer 3 Report

Dear Authors,

Thanks for all the corrections made. I accept them. I have only one technical note, previously overlooked: the title says "Clostridium Difficile infections..." and should be "Clostridium difficile infections...."

Best regards,

Reviwer